# Enhancing Sleep Quality: Assessing the Efficacy of a Fixed Combination of Linden, Hawthorn, Vitamin B1, and Melatonin

**DOI:** 10.3390/medsci12010002

**Published:** 2023-12-28

**Authors:** Matteo De Simone, Rosario De Feo, Anis Choucha, Elena Ciaglia, Francis Fezeu

**Affiliations:** 1Department of Medicine, Surgery and Dentistry “Scuola Medica Salernitana”, University of Salerno, Via Salvatore Allende, 84081 Baronissi, Italy; r.defeo9@studenti.unisa.it (R.D.F.); eciaglia@unisa.it (E.C.); 2Department of Neurosurgery, Aix Marseille University, APHM, UH Timone, 13005 Marseille, France; anis.c13@gmail.com; 3Department of Neurology & Neurological Surgery, Brain Global, 27659 Arabian Drive, Salisbury, MD 21801, USA; brainglobal.contact@gmail.com

**Keywords:** insomnia, sleep disorders, PSQI, ISI, DASS-21, linden, hawthorn, thiamine, melatonin, wearable smart watch

## Abstract

Sleep is essential for overall health, yet various sleep disorders disrupt normal sleep patterns, affecting duration, quality, and timing. This pilot study investigate the impact of a food supplement (SPINOFF^®^) on both sleep quality and mental well-being in 41 participants (mean age: 45.3 years). Initial assessments revealed sleep disturbances (Pittsburgh Sleep Quality Index—PSQ—mean score: 8.2) and insomnia symptoms (Insomnia Severity Index—ISI— mean score: 12.7). Mental health assessments showed psychological distress (Dass-21 Depression mean score: 4.2, Anxiety mean score: 6.9, Stress mean score: 11.6, Total mean score: 22.7). This study assessed sleep continuity using Awakenings per Night (ApN) via a smartwatch (HELO HEALTH^®^) and conducted the study in two phases: baseline (T0) and after 30 days of treatment (T1) (Phase A). No placebo-control was used in this study. After 30 days (Phase B), 21 patients were selected for reassessment. Eleven continued treatment for another 30 days (T2), while ten discontinued. Following the intervention, we observed remarkable improvements in sleep quality and mental distress. The SPINOFF^®^ supplement significantly reduced the PSQI scores (22.4%), indicating enhanced sleep quality. Additionally, there was a 19.6% decrease in ISI scores, demonstrating a reduction in insomnia symptoms. Moreover, overall psychological distress decreased by 19.5% signifying improved psychological well-being. In the second phase, participants who continued treatment experienced more substantial improvements, with a mean decrease of 0.8 points in PSQI scores (±0.9) and a mean decrease of 0.9 points in ISI scores. Our findings suggest that the SPINOFF^®^ supplement has the potential to effectively address both sleep disturbances and psychological distress in our study population.

## 1. Introduction

Sleep quality is of utmost importance for overall health and well-being; adequate sleep quality is essential for maintaining optimal physical health. During sleep, the body undergoes crucial restorative processes, such as tissue repair [1], hormone regulation, and immune system strengthening [2].

Sleep quality has a deep impact on mental health and emotional well-being [3]. Sufficient and restful sleep is crucial for cognitive functioning, memory consolidation, and emotional regulation [4].

High-quality sleep plays a vital role in optimal cognitive performance. It enhances concentration, attention, problem-solving abilities, creativity, and decision-making skills [5].

In addition, restorative sleep promotes the feeling of refreshed and rejuvenated energy, enabling people to engage in daily activities with vigor and enthusiasm [6]. Conversely, chronic poor sleep quality can negatively affect mood, with an increased risk of developing mental health disorders such as anxiety disorders, depression, and mood disturbances. It can diminish enjoyment of daily activities and reduce overall life satisfaction [7]. Poor sleep quality can impair cognitive function, leading to decreased productivity, reduced focus and memory problems [8]. Furthermore, extensive research data show an increased risk of various health conditions, including cardiovascular disease, obesity, diabetes, and impaired immune function [9,10]

Thus, in addition to the above-mentioned health problems that sleep disorders generate, they are a frequent cause of medical referral in a primary care setting [11], representing therefore also a social cost [12].

In the *Diagnostic and Statistical Manual of Mental Disorders, Fifth Edition* (DSM-5), insomnia disorder is defined as a persistent difficulty initiating or maintaining sleep, or experiencing non-restorative sleep, occurring at least three nights per week for a duration of at least three months. It is accompanied by significant distress or impairment in social, occupational, or other important areas of functioning. Insomnia disorder is further characterized by the presence of daytime symptoms such as fatigue, impaired concentration, mood disturbances, and reduced performance or productivity. The symptoms cannot be better explained by another sleep disorder, medical or mental health condition, or the effects of substance use [13].

It is useful to note that with DSM-5, the distinction between primary and secondary insomnia disappears, in favor of an umbrella category. The main reason for the removal of this distinction between primary and secondary insomnia was based on an NIH conference on insomnia in 2005 (National Institutes of Health, 2005), from the lack of evidence that treatment of the primary disorder, such as depression, alleviated insomnia as a result [14].

The most accepted model of the pathophysiology of insomnia is increased arousal, which is both the triggering and perpetrating substrate. So, this model explains both the onset and chronicity of the primitive sleep disorder [15,16]. This may reflect a dominance of arousal-generating brain areas over sleep-inducing brain areas [17]. Evidence supporting this main model focuses on finding that patients with insomnia have an increased power of fast electroencephalographic (EEG) frequencies during sleep with non-rapid eye movements [18].

However, like any occurrence in medicine, insomnia recognizes a very intricate pathogenesis. Certainly, genes are also involved, particularly certain polymorphisms located in genes coding for proteins involved in mental health (e.g., NPAS3), neuroplasticity (e.g., ROR1, PLCB1, EPHA4, and CACNA1A), neuronal excitability (e.g., GABRB1 and DLG2), and stress reactivity (e.g., STK39, USP25, and MARP10) [19].

In addition, the carrier status of polypoprotein E epsilon 4 (APOE-ε4) is an established risk factor for Alzheimer’s disease (AD) dementia. But it has also been linked to sleep disturbances in healthy older adults, and to an increased risk of insomnia [20].

Acute insomnia is very common, and does not need a specific treatment in many cases [21]. Chronic insomnia also has a significant prevalence [22], and is certainly a social burden [23]. Furthermore, this increasing prevalence is influenced as well by the spread of new entities such as post-COVID-19 syndrome (PCS) [24]. 

Primary care physicians generally prescribe these patients anxiolytics or hypnotics, usually some type of benzodiazepine [25], but only for short-term relief (maximum of 2–4 weeks). However, benzodiazepines and other hypnotics should be prescribed only with caution and for medically established conditions, as their use is not without risk [26,27]. Specifically, in the short term, benzodiazepines have a sedative effect, with memory loss, and have been accused of increasing hospitalization rates, disturbing balance in elderly subjects and reducing alertness in drivers [28]. In the long term, benzodiazepines can lead to dependence, with a withdrawal syndrome in about one-third of subjects after no more than 4 to 6 weeks of regular administration [29].

In this framework, especially in mild-to-moderate sleep disorders, a valuable contribution in regulating sleep–wake rhythm can come from the use of natural substance supplements [30]. This pilot study aims to investigate a supplement consisting of predetermined amounts of linden, hawthorn, vitamin B1 and melatonin in the treatment of mild/moderate insomnia. In addition, a literature review is proposed in relation to the effects of these four compounds. The main goal of the study is to evaluate, through validated questionnaires, the impact of the supplement SPINOFF^®^ on sleep quality and mental well-being in a cohort of 41 adults. A secondary purpose is to assess sleep continuity.

### 1.1. Linden

Linden (*Tilia* L., 1753) is a genus of trees or shrubs of the Malvaceae family. Several studies on the sedative and anxiolytic effects of various *Tilia* species have been carried out, in particular with *T. americana* L., *T. tomentosa* Moench, and *T. europaea* L. 

Similarly, the in vitro anxiolytic and sedative activities of *T. tomentosa*—the species present in the supplement studied in this paper—in hippocampal neurons has been reported, and data suggested that *T. tomentosa* extract binds to both GABAA and BDZ receptor sites [31]. 

### 1.2. Hawthorn

*Crataegus*, known as common hawthorn, is a genus of the family Rosaceae. It is native to Europe, Northwest Africa, and West Asia, but has been introduced in many other parts of the world. Hawthorn extracts have traditionally been used as an herbal remedy in complementary and alternative medicine. In folk phytotherapy, hawthorn is commonly used as a sedative, a hypotensive and for cardiovascular diseases [32]. Scientific evidence has demonstrated that *Crataegus* species are a source of nutrients, nutraceuticals, and bioactive compounds. The main bioactive compounds detected in hawthorn included epicatechin, hyperoside, rutin, vitexin, vitexin 2-O-rhamnoside, chlorogenic acid, hydroxycinnamic acid derivatives, proanthocyanidins, and anthocyanins and triterpenes, including botulinic, oleanolic and ursolic acids. Due to the action of tannins, these extracts promote a relaxing effect by increasing the feeling of well-being. Nowadays, hawthorn preparations are mostly used for the treatment of angina, hypertension, arrhythmias, congestive heart failure, and hyperlipidemia [33]. 

### 1.3. Vitamin B1

Vitamin B1, or thiamine, is one of the so-called water-soluble vitamins, those that cannot be accumulated in the body, but must be regularly ingested through food. Vitamin B1 plays an important role in energy metabolism, contributing to the important process of converting glucose into energy. The need for vitamin B1 varies according to gender: 1.2 mg per day for men and 0.9 mg for women, and can be met by following a normal diet. Vitamin B1 is required for the growth, development, and function of cells and for normal function of the brain, nerves and heart. For these reasons, it plays a very important role in the growth age of children [34].

In their study, Lee et al. mentioned a significant relationship between sleep duration and the dietary intake of certain micronutrients, with significant relation between sleep quality and most of micronutrients; they found that low thiamine intake was associated with oversleeping. However, short and long sleepers’ vitamin E, B1, and zinc intakes were found to be lower than those of normal sleepers, although retinol intake was found to be higher in short sleepers [35]. In contrast with this study [36], Grandner and colleagues found that vitamin B1 and E levels were lower in short sleepers, because B vitamins act in the regulation of the release of melatonin, which affects sleep. While there are a limited number of observational studies regarding the relationship between thiamine intake and long-sleep durations, overall previous studies have found no clear associations between low thiamine intake and sleep duration. A possible explanation is that there may be differences in susceptibility in varying populations. One mechanism that could explain oversleeping in association with low thiamine intake is that thiamine acts as a coenzyme for adenosine triphosphate (ATP) production. While ATP is a universal intracellular energy source in multiple tissues, sleep is suggested to be necessary for replenishing ATP in the nervous system. Conversely, sleep restriction inhibits this process, and thus decreases ATP levels in certain areas of the brain containing sleep-regulating neurons. Furthermore, the known hypnogenic properties of adenosine, the degradation product and ingredient of ATP, have been suggested to be mediators between lowered ATP levels and sleep induction, which further supports our explanation that low thiamine intake may lead to prolonged sleep [37]. However, recent studies have shown that thiamine can also act through non-coenzyme mechanisms, which may play an important role in neuroprotection in various pathologies. Thiamine and its derivatives have been shown to bind to mammalian proteins, including transcriptional regulator p53, poly(ADP-ribose) polymerase, prion protein PRNP, and a number of key metabolic enzymes that do not use ThDP as a coenzyme. Thiamine and its derivatives have also been shown to facilitate acetylcholinergic neurotransmission and activate synaptic ion currents. Although the exact molecular mechanisms of the non-coenzyme role of thiamine are not fully understood, further research in this area is required to identify proteins interacting with thiamine and its derivatives [38]. 

### 1.4. Melatonin

Melatonin is the main hormone involved in the control of the sleep–wake cycle. The synthesis of melatonin is regulated by the suprachiasmatic nucleus (SCN), and occurs in pineal gland: here, pineal cells first convert tryptophan into serotonin through hydroxylation and decarboxylation. N-acetyltransferase transforms serotonin into N-acetylserotonin, which is subsequently methylated by hydroxylindole-O-methyltransferase, to form melatonin. Melatonin mainly promotes sleep through its chronobiotic effects on the SCN. Melatonin acts on the CNS, promoting resynchronization when environmental conditions change. Elevated blood melatonin levels signal to tissues and organs that it is night-time, helping to regulate homeostasis. Melatonin resynchronizes the circadian rhythm and the sleep–wake cycle, and also regulates the reproductive cycle [39]. Melatonin levels decrease with age; thus, older adults are more prone to suffer from an inadequate melatonin level, and that is why reduced melatonin secretion may be involved in the mechanism of insomnia [40]. Melatonin and melatonin agonists play important roles in the treatment of insomnia, by activating MT1 and MT2 melatonin receptors. By activating these receptors, melatonin and non-selective MT1/MT2 receptor agonists have shown to improve sleep quality, increase total sleep time, improve sleep efficiency, and decrease sleep onset latency in insomnia patients. According to European guidelines, melatonin contributes to reduce sleep onset latency [41], which was also demonstrated for ramelteon [42]. Some of the original studies also investigated undesired side-effects, and concluded that melatonin is a safe drug. Although there has been an increase in interest in all aspects of sleep function/disorders, the knowledge about these topics is still incomplete: in fact, the effects were small, from a clinical point of view.

## 2. Materials and Methods

### 2.1. Study Design

This pilot study, conducted between April 2023 and September 2023, aimed to investigate the effects of a food supplement (SPINOFF^®^) on sleep quality and mental well-being, in a select group of participants. The study enrolled a total of 41 patients, all of whom were of Caucasian ethnicity and were chosen by general practitioners (GPs), in the Salerno area. Recruitment was facilitated through informal announcements, and participants underwent a medical assessment on the day of enrollment, with informed consent obtained after thorough information was provided.

The study population consisted of individuals aged 29 to 62, with a documented history of insomnia, in generally good health, and with a body mass index (BMI) falling within the range of 22.7 to 28.3 kg/m^2^. A critical factor for inclusion was that participants had refrained from using nicotine for at least the past 6 months. The diagnostic criteria for insomnia outlined in DSM-5 were applied for the selection of participants, along with screenings using validated questionnaires, including the Pittsburgh Sleep Quality Index (PSQI), Insomnia Severity Index (ISI), and the Depression Anxiety Stress Scales (DASS-21).

To be eligible for participation, individuals needed to score 5 or higher on the PSQI or 6 or higher on the ISI, indicating the presence of significant sleep disturbances. Participants were excluded from the study, based on the results of a medical examination, if they had an active infection, uncontrolled hypertension, major depression or an anxiety disorder as screened by the DASS-21, a history of cancer within the last 5 years, an unconventional sleep pattern, a diagnosed sleep disorder, or a chronic medical condition that could potentially affect energy or fatigue levels. Exclusion criteria also applied to those currently experiencing a major depressive episode (as determined by their DASS-21 score), individuals with allergies to the study products, those who had consumed more than 400 mg of caffeine per day within the previous 2 weeks, individuals who had used psychotropic medications, stimulants, cannabis, non-registered drug products, or illicit substances in the preceding 4 weeks, those at risk of drug or alcohol abuse, and participants who had used any sleep aids in the past 2 weeks. Lastly, pregnant women were also excluded from the study.

The study was conducted in accordance with the Declaration of Helsinki, and, as indicated in the 2015 ministerial guidelines (Ministero della Salute direzione generale per l’igiene, la sicurezza degli alimenti e la nutrizione—ufficio IV ex dgsan. Linee di indirizzo sugli studi condotti per valutare la sicurezza e le proprietà di prodotti alimentari), inspired by the indications of the E.F.S.A. (European Food Safety Authority), the approval of an ethical committee is not required [43].

A total of 56 potential participants were selected, whom GPs had suggested should take the supplement; however, those who agreed and completed 4 weeks’ treatment, numbered 41. Participants were advised to receive a food supplement based on linden, tilia tomentosa (150 mg e.s.), hawthorn, *C. oxyacantha* (150 mg e.s.), vitamin B1 (1.65 mg, 150% VNR), and melatonin (1 mg), in the form of bucky tablets (SPINOFF^®^ Agaton Srl). The selection of these components is based on their potential sleep-enhancing properties and previous research indicating their efficacy in improving sleep quality. Linden promotes relaxation (sleep in cases of stress). The beneficial effect is obtained from the intake, just before bedtime. For a more detailed analysis, refer to the previous section.

The dosing regimen and duration of treatment was determined based on previous studies and clinical experience. Participants were instructed to take one capsule daily, before bedtime, for 4 weeks. The dosage and duration was e carefully chosen to provide a reasonable trial period for assessing the intervention’s effects, while minimizing potential side effects.

The primary outcome was about sleep quality. This outcome was assessed using validated sleep questionnaires, PSQI and ISI [44]. 

Patients deciding they wanted to take the oral supplement were asked to fill out three questionnaires. Two questionnaires were used to assess the extent of sleep disorders, and these were the PSQI and ISI. The PSQI is a validated questionnaire; the PSQI global score > 5 has a sensitivity of 98.7 and a specificity of 84.4 as a marker of sleep disorders in patients with insomnia, compared with controls [45]. The ISI also has adequate internal consistency, and is a reliable self-report measure to assess perceived sleep difficulties [46]. Finally, the third questionnaire used was the DASS-21, which expresses the measure of general distress articulated in three additional orthogonal dimensions (anxiety, depression and stress) [47].The choice of these three questionnaires lies in the usefulness of framing the sleep disorders of individual patients and studying their sleep-related distress. 

These questionnaires provide comprehensive assessments of sleep quality, duration, disturbances, and daytime dysfunction. We structured our study into two phases: Phase A and Phase B (Figure 1). Specifically, these questionnaires were administered at time zero (T0) and after 30 days of usage of the study product (T1). At the time of the 30-day evaluation (at T1), 21 patients were selected. A total of 11 of these continued taking the oral supplement for an additional 30 days, and new consent was collected to reevaluate them (with PSQI and ISI) at the end of this period (T2). The other 10, because they had achieved clinically significant improvement, discontinued the intake (T2c); they were evaluated 30 days after discontinuation, with both PSQI and ISI questionnaires, as an internal control group [48]. 

In addition to the primary outcome, we also considered the secondary outcome of the number of awakenings per night. This additional parameter provides valuable insights into the overall sleep pattern, and contributes to a comprehensive assessment of the effects of the interventions [49]. By including the number of awakenings per night (ApN) as a secondary outcome, we aimed to capture a more detailed picture of the potential benefits of the studied intervention on sleep continuity and disturbances throughout the night.

This parameter was evaluated in two different nights (the night before the T0 and T1 evaluations), with the help of a wearable smart watch (LifeWatch Generation 2) from Helo Health, a data-driven wellness technology company. For any further information, contact the manufacturer at the website https://helohealth.com/about-us (accessed on 28 September 2023).

### 2.2. Statistical Analysis

All the statistical analyses were performed using the Statistical Platform R, vers. 4.1.2. (R Core Team (2021); R: a language and environment for statistical computing, the R Foundation for Statistical Computing, Vienna, Austria). Normality of numerical variables was assessed using the Shapiro–Wilk test. Standard descriptive statistics were used to characterize the cohort: numerical variables were described using mean ± standard deviation (std. dev.) with the range and categorical factors synthesized using absolute frequencies and percentages. The significance of the within-group differences (Phase A) were assessed using the *t*-test for paired samples, both for absolute and percentage variations, whereas between-group differences (Phase B) were evaluated using the *t*-test for unpaired samples. To explore the possible association between clinical and demographic characteristics and the observed variation in the outcome variables, multivariate linear regression models were estimated; in these models, the change from baseline was entered as the dependent variable and age, gender and BMI were entered as independent variables. All these models were adjusted with respect to the baseline values of the corresponding outcome variable.

### 2.3. Demographic and Clinical Characteristics of Patients 

In this study, a total of 41 patients were included, with a mean age of 45.3 years (±7.8, ranging from 29 to 62 years). Among the participants, 56.1% were male. The average Body Mass Index (BMI) in this sample was 26.2 kg/m^2^ (±1.4, ranging from 22.7 to 28.3 kg/m^2^), reflecting a diverse demographic mix.

Sleep quality and insomnia severity were assessed using the Pittsburgh Sleep Quality Index (PSQI) and the Insomnia Severity Index (ISI), respectively. The participants had an average PSQI score of 8.2 (±1.9, ranging from 5 to 13), indicating initial sleep disturbances. Regarding insomnia severity, the mean ISI score was 12.7 (±3.6, ranging from 6 to 21). Within this context, 7.3% of patients had no clinically significant insomnia (0–7 ISI score), while 63.4% exhibited subthreshold insomnia (8–14 ISI score), and 29.3% experienced moderate clinical insomnia (15–21 ISI score).

Furthermore, participants’ psychological well-being was evaluated using the Depression, Anxiety, and Stress Scales-21 (Dass-21). The mean Dass-21 Depression score was 4.2 (±2.4, ranging from 0 to 9), the Dass-21 Anxiety score was 6.9 (±2.4, ranging from 2 to 11), and the Dass-21 Stress score was 11.6 (±3.4, ranging from 4 to 20). The overall psychological distress, indicated by the Dass-21 Total score, averaged at 22.7 (±5.9, ranging from 14 to 37).

Lastly, the assessment of sleep continuity revealed that patients experienced a median of 3 ApN, with an interquartile range of [2; 3], ranging from 0 to 6 awakenings. These demographic and clinical characteristics provide valuable insights into the patient population under study, highlighting the heterogeneity in sleep quality, insomnia severity, and psychological well-being among the participants. For all the data, refer to Table 1.

Moreover, clinical characteristics, including the presence of medical comorbidities, and any relevant concurrent treatments, were documented. This insight into the demographic and clinical profile of the patient population establishes a foundation for the subsequent interpretation of the study results and their potential implications for sleep management strategies. 

## 3. Results

Changes in the score of the two scales used to frame patients’ sleep disturbance, PSQI and ISI, were respectively analyzed. Findings of the change between baseline status (at the time of clinical consultation) and after one month of treatment of the Dass-21 scale are also reported. 

Table 2 contains the assessment of the statistical significance of the variations found on the clinical scales, both in absolute and percentage terms.

The study’s results indicate significant improvements in various sleep and mental health-related parameters among the participants. In terms of sleep quality, as assessed by the Pittsburgh Sleep Quality Index (PSQI), there was a substantial absolute reduction of −1.8 points (95% CI: −2.2 to −1.4), with a *p*-value of <0.001. This reduction corresponds to a 22.4% improvement (95% CI: −27.5 to −17.3) in sleep quality. Additionally, the Insomnia Severity Index (ISI) demonstrated a significant absolute reduction of −6.3 points (95% CI: −7.2 to −5.4) with a *p*-value of <0.001, indicating a 19.6% decrease (95% CI: −24.5 to −14.7) in insomnia severity.

Regarding mental health, the study found a reduction in anxiety and stress levels. The Dass-21 Anxiety score showed an absolute reduction of −0.9 points (95% CI: −1.3 to −0.5), with a *p*-value <0.001, representing a 12.7 percent (95% CI: −17.9 to −7.5) decrease in anxiety symptoms. Similarly, the Dass-21 Stress score showed a significant absolute reduction of −1.1 points (95% CI: −1.6 to −0.6), with a *p*-value of <0.001, signifying a 7.6% (95% CI: −13 to −2.1) reduction in stress levels. In contrast, Dass-21 Depression showed no statistically significant changes following supplement intake. In particular, Dass-21 Depression score showed a non-significant absolute reduction of −0.1 points (95% CI: −0.5 to 0.2), with a *p*-value of 0.412. The percentage change in depression symptoms was 4.4% (95% CI: −8.4 to 17.1), but did not reach statistical significance.

Overall psychological distress, as assessed by the Dass-21 Total score, displayed a substantial absolute reduction of −4.9 points (95% CI: −6 to −3.9), with a *p*-value of <0.001, indicating a 19.5% decrease (95% CI: −23.4 to −15.6) in overall psychological distress.

Additionally, the study investigated ApN as a secondary outcome, showing a significant absolute reduction of −1 point (95% CI: −1.5 to −0.5), with a *p*-value of <0.001, suggesting improvements in sleep continuity.

During Phase B, we evaluated changes in PSQI and ISI scores. Our results, presented in Table 3, reveal significant differences in these sleep-related variables between the two groups. Specifically, subjects who discontinued treatment showed a mean decrease of 0.7 points in PSQI scores (±0.7), while those who continued treatment showed a mean decrease of 0.8 points in PSQI scores (±0.9). The mean difference between the two groups was −1.5 (95% CI: −2.2 to −0.8), with a *p*-value less than 0.001 (indicating a significantly greater decrease in PSQI scores in the group that did not discontinue treatment). In addition, for the Insomnia Severity Index (ISI), subjects who discontinued treatment showed a mean decrease of 0.6 points (±1.4), while subjects who maintained treatment showed a mean decrease of 0.9 points (±1.3). The mean difference for ISI scores was −2 (95% CI: −3.2 to −0.7), with a *p*-value of 0.004, showing a significant difference in favor of maintaining treatment.

## 4. Discussion

Insomnia is a widespread issue, with a significant impact on individuals’ well-being. According to the National Sleep Foundation, approximately 30% of adults experience short-term insomnia, while around 10% suffer from chronic insomnia disorder, which can be particularly prevalent among postmenopausal women [50]. While conventional treatments for insomnia often involve sedative drugs, there is growing concern about the misuse of these medications and the risk of developing dependencies. The potential long-term consequences of relying on such drugs have led to a search for alternative treatments. In this context, there is a rising interest in the use of vitamins to address sleep disorders, which has also given rise to modern controversies.

Although medication is the primary treatment for disabling insomnia, it is critical to recognize the effectiveness of cognitive behavioral therapy for insomnia (CBT-I), in managing sleep disorders. As shown in a metanalysis by Trauer et al., CBT-I has been shown to be effective in improving sleep quality and addressing the underlying causes of insomnia. In particular, an improvement was observed concerning sleep onset latency (SOL), wakefulness after sleep onset (WASO), total sleep time (TST) and sleep efficiency [51].

However, the availability of health care providers who specialize in CBT-I must be taken into account. In situations where access to qualified CBT-I providers is limited, medication may indeed be a more viable option. However, in cases where both options are accessible, a personalized approach that takes into account patient preferences, severity of insomnia, and potential contraindications should guide the choice between medication and CBT-I [52].

Many studies have used oral supplements, particularly those with melatonin, magnesium, B-complex vitamins, vitamin E, and vitamin D, to treat insomnia. 

Before reviewing these clinical studies, we should actually say that evidence on the mechanism of action often comes from preclinical studies; for example, it has been shown in rats that thiamine has a function as a modulator of many diurnal rhythms, such as body temperature and sleep–wake rhythm. In fact, vitamin B1 regulates the latter by influencing the efficiency of GDH, thus altering brain glutamate levels, depending on the timing of its supplementation [53,54]. This information could guide clinicians in choosing the optimal timing of supplement administration; however, more evidence is needed, especially for supplements that clearly contain more than one molecule.

Studies have used the PSQI and ISI as tools to assess sleep disorders in comorbidity, with many different clinical conditions, such as menopause, traumatic brain injury, cancer, and cardiovascular disease, showing that we are facing a hot and current problem. Chan and Lo, in 2022, collected all these studies by publishing a meta-analysis on the efficacy of these supplements, stratifying them by patient cohort [30].

One study involving 160 postmenopausal women with chronic insomnia disorders sought to investigate the effectiveness of vitamin E as a treatment option, finding that the group with vitamin E showed significant improvement in sleep quality [55].

An RCT by Grima and colleagues studied the use of 2 g of melatonin in 33 patients with moderate to severe TBI, who developed insomnia as a result of trauma, and they observed a significant improvement in sleep quality [56]. Ostadmohammadi et al. demonstrated, in a cohort of 60 hemodialysis patients, that melatonin supplementation for 12 weeks had beneficial effects on mental health, glycemic control, inflammatory markers and oxidative stress [57].

The originality of our results are not only highlighted in the clinical improvement in overall sleep that patients record at the end of the treatment period with the oral supplement, but also in the improvement in general well-being. In fact, in this study, it was possible to confirm not only the correlation of the clinical assessment with the already validated questionnaires for sleep disorders, but also a correlation with the global distress generated by the disorder itself, and its improvement, congruent with the improvement in the sleep quality.

The pilot study involved a total of 41 subjects who came to GPs’ outpatient clinics reporting sleep disorders. These patients, according to the latest available guidelines on the treatment of insomnia [13], and based on evidence-based clinical practices, are not intended, at least at first blush, to be treated with drugs such as benziodiazepines and z-drugs. The first approach should be based on sleep hygiene and the use of supplements based on natural molecules. Therefore, it was suggested that they take a dietary supplement (a fixed combination of linden, hawthorn, vitamin B1 and melatonin: SPINOFF^®^). This study did not include a control group, because in our opinion it would have been unethical to leave some of these patients, who reported sleep disorders, without any treatment. Ethical problems of a control group were shown in multiple studies. As early as 2001, E. J. Emanuel and F. G. Miller critically reviewed the ethical problems about the use of control groups, noting their only partial compliance with the latest version of the Declaration of Helsinki [58]. Another major study reviewed the methodological justifications for including a control group. None of the justifications identified by those authors may be suitable to suggest the use of a control group for this study [59]. Countless other studies have highlighted and supported this evidence [60,61,62]. The absence of a control group, driven by ethical considerations, has made it challenging to distinguish the true therapeutic effects of the drug from the placebo response. While our findings indicate positive outcomes, it is essential to acknowledge the inherent limitations in interpreting these results and the need for further research that incorporates control groups, to better understand the true efficacy of the treatment. This consideration highlights the delicate balance between ethical concerns and scientific rigor in clinical research.

Evaluation at T0 reveals that participants had a mean PSQI score of 8.2 (±1.9, ranging from 5 to 13), and a mean ISI score of 12.7 (±3.6, ranging from 6 to 21), suggesting early sleep disorders. At T0, psychological well-being was assessed using the Depression, Anxiety, and Stress Scales-21 (Dass-21). The mean Dass-21 Depression score was 4.2 (±2.4, 0 to 9), Dass-21 Anxiety score was 6.9 (±2.4, 2 to 11), and Dass-21 Stress score was 11.6 (±3.4, 4 to 20). Overall psychological distress, indicated by the total Dass-21 score, averaged 22.7 (±5.9, 14 to 37). Finally, assessment of sleep continuity revealed that patients experienced a median of three ApN, with an interquartile range of [2; 3], ranging from 0 to 6 awakenings.

Participants then took an oral supplement for 30 days, and they were re-evaluated with the same questionnaires, as planned in the study protocol.

The results at T1 show a significant improvement in the sleep quality of the patients, evidenced in a superimposable way, by the improvement in both scores of the two questionnaires and the ApN. Specifically in relation to PSQI, there was an absolute reduction of −1.8 points, with a *p* value < 0.001. This reduction corresponds to a 22.4% improvement in sleep quality. Comparably, the ISI demonstrated a significant reduction of −6.3 points, with a *p*-value of <0.001, indicating a 19.6% decrease in insomnia severity. Figure 2 shows instead the effect of demographic characteristics (age, gender and BMI) on changes in scores following supplementation, obtained through linear regression models. It emerges that in males there is a smaller reduction on the ISI and PSQI (not significant) scale than in females, but a greater reduction on the Depression subscale of the DASS-21; patients with high BMI also benefited more in terms of reduction in the anxiety score. No significant correlations are shown between the different subscales (in terms of changes).

These results could be incidental findings, due to the small sample size analyzed. However, the statistical significance of the greater improvement among the male sex on the ISI questionnaire (*p* = 0.027) is congruent with that of the PSQI questionnaire, although it does not reach statistical significance (*p* = 0.101). Although this result, to the best of our knowledge, is not described in the current literature, it could be explained as a different vulnerability between males and females to the placebo effect [63].

The study’s Phase B evaluation, examining changes in PSQI and ISI scores, underscores the importance of treatment continuity. Subjects who continued treatment displayed greater improvements in sleep quality, as evidenced by a mean decrease of 0.8 points in PSQI scores, compared to those who discontinued treatment, with a mean decrease of 0.7 points. A significant mean difference of −1.5 in favor of maintaining treatment for PSQI scores highlights the benefits of treatment continuity. Similarly, for the Insomnia Severity Index (ISI), subjects who maintained treatment exhibited superior outcomes, with a mean decrease of 0.9 points, compared to 0.6 points in those who discontinued treatment. This statistically significant difference, with a mean difference of −2, further underscores the importance of continuous treatment in managing insomnia severity.

Future research in the field of insomnia should explore innovative interventions and personalized treatment approaches tailored to individual sleep profiles. Investigating the potential benefits of emerging technologies, such as wearable sleep-monitoring devices [64] and telehealth interventions [65], could yield valuable insights into more effective insomnia management. 

Furthermore, longitudinal studies tracking the long-term effects of insomnia treatments and their impact on mental and physical health outcomes are crucial. Understanding the relationships between insomnia, comorbid conditions, and overall quality of life can inform holistic treatment strategies.

Additionally, research should delve deeper into identifying and addressing the underlying causes of insomnia, including psychosocial factors, genetics, and circadian rhythm disruptions. Targeted interventions addressing these root causes may lead to more sustainable and personalized insomnia treatments.

Lastly, the development of evidence-based guidelines for insomnia management that incorporate the latest research findings is essential for informing clinical practice and improving patient outcomes. Overall, future research should aim to enhance our understanding of insomnia’s multifaceted nature and refine treatment strategies, to provide better support for individuals struggling with sleep disturbances.

## 5. Limitations

The pilot study’s small sample size and lack of an external control group may limit the generalizability of the findings. The absence of a proper control group makes it challenging to differentiate the effects of the intervention from other confounding factors or the placebo effect.

The open-label design may introduce bias in subjective outcomes, as participants and researchers are aware of the treatment being administered. The lack of blinding may influence participants’ perceptions and expectations of the intervention’s effects. Considerations of longer-term effects may not be fully captured within the short duration of the study. The pilot study aims to provide initial insights into the intervention’s safety and the efficacy of the food supplement, but a longer-term study would be required to evaluate the intervention’s sustained effects and long-term safety profile.

## 6. Conclusions

This pilot study explored the potential of a supplement consisting of linden, hawthorn, vitamin B1, and melatonin in the management of mild-to-moderate insomnia. The study revealed a supplement’s positive impact on various aspects of sleep quality and psychological well-being.

The participants in this study demonstrated a remarkable improvement in sleep quality, as evidenced by substantial reductions in the Pittsburgh Sleep Quality Index (PSQI) and Insomnia Severity Index (ISI) scores. These findings suggest that the supplement effectively addresses sleep disturbances, leading to enhanced sleep continuity and reduced insomnia severity.

While the observed reduction in depression scores did not reach statistical significance, the overall psychological distress, as measured by the Dass-21 Total score, demonstrated a notable decrease. These results indicate that the supplement has a positive impact on psychological well-being, which can be instrumental in managing sleep-related issues.

Notably, the participants who continued taking the supplement in Phase B increased their improved sleep quality, and reduced their insomnia severity. This underscores the supplement’s potential for long-term benefits in sleep management.

While the mechanisms underlying these positive effects warrant further investigation, this study offers promising insights into the role of linden, hawthorn, vitamin B1, and melatonin in promoting sleep quality and overall well-being. The supplement may provide a valuable non-pharmacological approach to managing mild-to-moderate sleep disorders as required by guidelines, potentially reducing the need for short-term anxiolytics or hypnotics.

In a world where sleep disorders continue to impact the physical and mental health of many individuals, this supplement and the device used to assess the results offers a promising avenue for improving sleep quality and fostering better psychological well-being. Further research and clinical trials are needed to validate these findings and elucidate the supplement’s full potential. Nevertheless, this study contributes to the growing body of knowledge on non-pharmacological approaches to managing sleep disorders and promoting a healthier, more restful sleep experience.

## Figures and Tables

**Figure 1 medsci-12-00002-f001:**
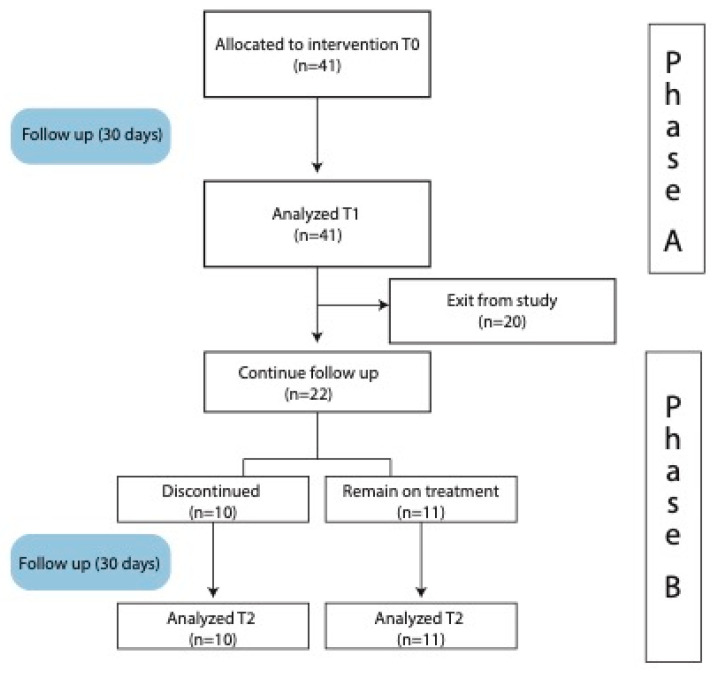
Flow chart of the study design (Phase A and B).

**Figure 2 medsci-12-00002-f002:**
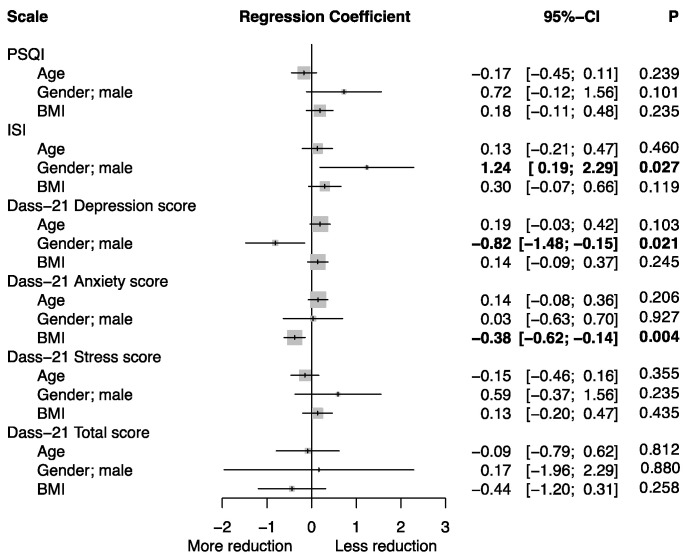
Associations between Age, Gender and BMI and the mean change from baseline for the different outcome variables. Results are expressed as regression coefficients—estimated using multivariate linear regression—with the corresponding 95% Confidence Intervals (95% CI).

**Table 1 medsci-12-00002-t001:** Baseline demographics and clinical characteristics of the study population.

Variable	Overall Sample (*n* = 41)
Age; years	45.3 ± 7.8 (29 to 62)
Gender; male	23 (56.1)
BMI; (kg/m^2^)	26.2 ± 1.4 (22.7 to 28.3)
PSQI (Pittsburgh Sleep Quality Index)	8.2 ± 1.9 (5 to 13)
ISI (Insomnia Severity Index)	12.7 ± 3.6 (6 to 21)
No clinically significant insomnia (0–7)	3 (7.3)
Subthreshold insomnia (8–14)	26 (63.4)
Moderate clinical insomnia (15–21)	12 (29.3)
Dass-21 Depression score	4.2 ± 2.4 (0 to 9)
Dass-21 Anxiety score	6.9 ± 2.4 (2 to 11)
Dass-21 Stress score	11.6 ± 3.4 (4 to 20)
Dass-21 Total score	22.7 ± 5.9 (14 to 37)
HpN	3 [2; 3] (0 to 6)

Variables are described as mean ± standard deviation (min to max), median [1st; 3rd quartile] (min to max) and absolute frequencies, with percentages.

**Table 2 medsci-12-00002-t002:** Mean change from baseline, with the corresponding 95% Confidence Intervals in the overall sample. Absolute variations refer to the difference between end of follow-up and baseline values, while percentage variations were obtained by dividing absolute change by baseline values.

Variable	Mean (95% CI) (*n* = 41)	*p*
PSQI (Pittsburgh Sleep Quality Index)		
Absolute	−1.8 (−2.2 to −1.4)	<0.001
Percentage	−22.4 (−27.5 to −17.3)	<0.001
ISI (Insomnia Severity Index)		
Absolute	−6.3 (−7.2 to −5.4)	<0.001
Percentage	−19.6 (−24.5 to −14.7)	<0.001
Dass-21 Depression score		
Absolute	−0.1 (−0.5 to 0.2)	0.412
Percentage	4.4 (−8.4 to 17.1)	0.493
Dass-21 Anxiety score		
Absolute	−0.9 (−1.3 to −0.5)	<0.001
Percentage	−12.7 (−17.9 to −7.5)	<0.001
Dass-21 Stress score		
Absolute	−1.1 (−1.6 to −0.6)	<0.001
Percentage	−7.6 (−13 to −2.1)	0.008
Dass-21 Total score		
Absolute	−4.9 (−6 to −3.9)	<0.001
Percentage	−19.5 (−23.4 to −15.6)	<0.001
Hpn		
Absolute	−1 (−1.5 to −0.5)	<0.001

**Table 3 medsci-12-00002-t003:** Analysis of change during Phase B in subjects who discontinued and in subjects who remained in treatment after the end of Phase A.

Variable	Discontinued (*n* = 10)	On Treatment (*n* = 11)	Mean Difference (95% CI) *	*p*
PSQI (Pittsburgh Sleep Quality Index)	0.7 ± 0.7	−0.8 ± 0.9	−1.5 (−2.2 to −0.8)	<0.001
ISI (Insomnia Severity Index)	0.6 ± 1.4	−0.9 ± 1.3	−2 (−3.2 to −0.7)	0.004

* Adjusted for the corresponding outcome variable, as measured at the end of Phase A.

## Data Availability

The data presented in this study are available on request from the corresponding author. The data is not publicly available due to privacy reasons.

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
