# Peer review of "Enhancing Sleep Quality: Assessing the Efficacy of a Fixed Combination of Linden, Hawthorn, Vitamin B1, and Melatonin"

_medsci, 2023, doi:10.3390/medsci12010002_

Round 1

Reviewer 1 Report

Comments and Suggestions for Authors

Comments to the paper

1) The ‘Abstract’ requires redrafting and clarification (e.g. too little space is devoted to specific research results, variable ranges seem unnecessary, SD is enough, etc.);

2) The objective should be specified in the ‘Introduction’ (especially, please indicate which group the study concerned and the period of using the supplement);

3) Please clearly state the inclusion criteria as you did for the exclusion criteria;

4) Lacking titles for Figures 1 and 2;

5) Lack of characteristics regarding the statistical analysis used, please add (statistical package, controlling normality of distribution, statistical tests, assumed level of significance, etc.);

6) There are no reliability indicators of the research tools used in the ‘Methods’ section, please add;

7) Please, check the consistency of data regarding the analysed variables in individual sections of the work (including age, BMI, e.g. in section 2.1 the age of the participants is 25-65, and in section 2.2 29-62 years, similarly for BMI - there are divergent results). Please explain and correct;

8) In Table 1, I suggest writing M and SD as M ± SD (not +-);

9) The title of Table 2 should be clarified (to make it readable without analysing the entire text);

10) The ‘Discussion’ section is actually a summary of the work (methodology and results), but has little to do with the discussion of the results. This section requires thorough revision in order to outline regularities and trends from your own research, interpret them and then refer them to other research in a similar area. In this approach, the ‘Discussion’ simply does not exhaust the features of this section of the work;

11) In the analyses individual characteristics (age, gender, BMI, etc.) were not taken into account, which also seems to be a limitation of the work;

12) The ‘Conclusions’ require thorough correction, because in the current version, they are once again another summary of the work. ‘Conclusions’ are intended to directly respond to the purpose of the research (and the research questions developing it);

13) Directions for future research should be at the end of the ‘Discussion’ or in the ‘Limitations’, not in the ‘Conclusions’.

Author Response

1) The ‘Abstract’ requires redrafting and clarification

We thank the reviewer for noticing this. These part is now improved in the revised version of the paper.

2) The objective should be specified in the ‘Introduction’ 

We thank the reviewer for noticing this. These part is now improved in the revised version of the paper.

This pilot study aims to investigate a supplement consisting of predetermined amounts of linden, hawthorn, vitamin B1 and melatonin in the treatment of mild/moderate insomnia. In addition, a literature review is proposed in relation to the effects of these four compounds. In particular the study aimed to investigate the impact of the food supplement SPINOFF® on sleep quality and mental well-being in a group of 41 participants, with a mean age of 45.3 years. The study addressed initial sleep disturbances via PSQI and ISI and mental health via DASS-21. The study also assessed sleep continuity using Awakenings per Night (ApN) via a smartwatch (HELO HEALTH®) and was conducted in two phases: baseline (T0) and after 30 days of treatment (T1) (Phase A). In Phase B, 21 patients were reassessed, with 11 continuing treatment for another 30 days (T2), and ten discontinuing.

3) Please clearly state the inclusion criteria as you did for the exclusion criteria We thank the reviewer for noticing this. These part is now improved in the revised version of the paper.

4) Lacking titles for Figures 1 and 2;

We thank the reviewer for noticing this. These titles were now added in the revised version of the paper.

Figure 1: Flow chart of the study design

Figure 2: Associations between Age, Gender and BMI and the mean change from baseline for the different outcome variables. Results are expressed as regression coefficients estimated using multivariable linear regression with the corresponding 95% Confidence Intervals.

5) Lack of characteristics regarding the statistical analysis used, please add (statistical package, controlling normality of distribution, statistical tests, assumed level of significance, etc.);

We thank the reviewer for noticing this. The following paragraph was added in the revised version of the paper.

All the statistical analyses were performed using the Statistical Platform R, vers. 4.1.2. (R Core Team (2021. R: A language and environment for statistical computing. R Foundation for Statistical Computing, Vienna, Austria.). Standard descriptive statistics were used to characterize the cohort: numerical variables were described using mean ± standard deviation (std. dev.) with range and categorical factors were synthesized using absolute frequencies and percentages. The significance of the within-group differences (Phase A) were assessed using t-test for paired samples both for absolute and percentage variations whereas between group differences (Phase B) were evaluated using t-test for unpaired sample. To explore the possible association between clinical and demographical characteristics and  the observed variation in the outcome variables, multivariable linear regression models were estimated; in these models the change from baseline entered as the dependent variables and  age, gender and BMI entered as independent variable. All these models were adjusted by the baseline values of the corresponding outcome variable. In all the analyses, P values < 0.05 were deemed as statistically significant

6) There are no reliability indicators of the research tools used in the ‘Methods’ section, please add;

We thank the reviewer for noticing this. These part is now improved in the revised version of the paper.

7) Please, check the consistency of data regarding the analysed variables in individual sections of the work

We thank the reviewer for noticing this. These part is now improved in the revised version of the paper.

8) In Table 1, I suggest writing M and SD as M ± SD (not +-);

The content of Table 1 was changed accordingly to the reviewer suggestion

9) The title of Table 2 should be clarified (to make it readable without analysing the entire text);

The title of Table 2 was changed accordingly to the reviewer suggestion

10)We thank the reviewer for noticing this. These part is now improved in the revised version of the paper.

11) In the analyses individual characteristics (age, gender, BMI, etc.) were not taken into account, which also seems to be a limitation of the work;

We thank the reviewer for this remark. Actually, the effect of age, gender and BMI on the observed variations was estimated using multivariable linear regression and the results were reported in figure 4. We added in the statistical analysis section more details on this analysis.

12) The ‘Conclusions’ require thorough correction, because in the current version, they are once again another summary of the work. ‘Conclusions’ are intended to directly respond to the purpose of the research

We thank the reviewer for noticing this. These part is now improved in the revised version of the paper.

13) Directions for future research should be at the end of the ‘Discussion’ or in the ‘Limitations’, not in the ‘Conclusions’.

We thank the reviewer for noticing this. These part is now improved in the revised version of the paper.

Reviewer 2 Report

Comments and Suggestions for Authors

The paper by Matteo De Simone and colleagues caught my attention, and I believe, the topic is interesting for the public too. Unfortunately, while careful reading of the section describing the experimental design, I realized, the design is wrong. The paper has no placebo group, thus the results can not be assigned to the effect of the pills. Instead, they could result from the behavior of the patients (you may sleep better just because you are included in a sleep study) or because of the well-known placebo effect. The authors have to repeat the study with the placebo group included.

Before realizing the problem of the experimental design, I've made some comments to the abstract, title and introduction. I'll keep them below just in case.

_________________________________________________________________________

The title should be shortened. I'd suggest choosing, whether to use the words "Enhancing Sleep Quality" or "insomnia and distress", but not both. You should also delete the phrase "a review and a pilot study". 

The abstract should be shortened to about 250 words, but the sentences like "Following the intervention, the study observed significant improvements in sleep quality and mental distress." (line 30) should include more details. Particularly, your description of the parameters within the groups (lines 18-25) is written in a good style, so here you should keep it. Include some details about the "significant improvements".

The last sentence of the abstract should be deleted. This paper is not a review.

Include the aim of this study in the abstract.

The general problem of this manuscript is, that it's too raw. This also makes it difficult to review the paper, thus I have to give some general recommendations for the revision.

Introduction to any paper should be of 1-1.5 papers. Of course, providing information about each component (now organized as 1.1-1.4) is important for the study, as well as the references (and you should use more of them), but many sentences in the Introduction should be shortened or excluded. For example, there is no use in the duplication of sentences in lines 39 and 43; the text in the paragraph in lines 61-69 is too long - you can just use this information and cite the book, etc.

Methods. You should cite the references, when making statements such as "Linden promotes 271 relaxation (sleep in case of stress)" etc.
Could you explain the chosen time of taking the tablets? Diurnal regulation is known at least for both melatonin and thiamine, so do they support each other from this point of view?

A clear paragraph on statistics should be added.

The figure captions are missing, while they are very important and should explain the figures.

Comments on the Quality of English Language

Unfortunately, the paper has no placebo group, thus the results can not be assigned to the effect of the pills. Instead, they could result from the behavior of the patients (you may sleep better just because you are included in a sleep study) or because of the well-known placebo effect.
The authors have to repeat the study with the placebo group included. This would also increase the statistics.
The quality of the manuscript is very low too.

Author Response

Dear reviewer,
Thank you for your review; we hope we have understood it in its desirable one constructive vein.
As you will read in the manuscript it is made clear that the inclusion of a true control group is precluded on ethical grounds, which we are sure you can understand. Second, the intellectual honesty of these authors made us include in the limitations this hypothetical bias that you mention. What's more, as you may have read from the study design, an internal control group is included at time T2: specifically, a group of patients who had previously gained improvement from supplement use and who had discontinued it voluntarily had worsened their sleep quality again; the group who had continued the intake, on the other hand, had maintained if not further improved their sleep quality.
In this regard, we have added as a reference a study from the NEJM, which supports what we are saying: the size and quality of the internal control group thus structured is adequate to attribute the observed effects to the supplement and not to the placebo effect.

In relation to the other points you noted, you will be able to see that the manuscript has been revised and corrected, following your instructions. In particular, the title the abstract the introduction, the materials and methods section, the discussion, and the conclusion have been changed.

Reviewer 3 Report

Comments and Suggestions for Authors

Author Response

DEAR REVIEWER,

Thank you for this report and below we hope to provide you with appropriate and relevant answers on each point.

In particular, we would like to bring to your attention two key aspects that are, nevertheless, elaborated on below. Attached you will find the ministerial guidelines governing the use of supplements and exempting us from the ethics committee requirement. Moreover, as you will have also noted in the discussion of the paper we cite the EFSA guidelines from which our ministry of health is guided.
The other aspect concerns the control group, our ethical choice does not conflict with the statistical power of the study since several works have shown how having an internal control group that discontinues the treatment and is then evaluated is an equally robust way to study the effect of a supplement while avoiding a placebo effect. So as suggested by the other reviewers all these aspects are properly clarified to avoid any misunderstanding.

General 

1) We appreciate the reviewer's observation regarding the need for a clearer distinction between sleep quality and sleep continuity in our study. In response to this feedback, we will revise the manuscript to provide explicit definitions of these terms and their respective measurements within the context of our study. We recognize that the use of the Insomnia Severity Index (ISI) may introduce ambiguity, as it encompasses both sleep quality and continuity aspects. To enhance clarity, we will explicitly delineate how our study specifically addresses and measures each component, providing a more nuanced understanding for readers. We value the reviewer's input and will ensure that the revised manuscript provides a more detailed and precise description of the distinctions between sleep quality and sleep continuity, particularly in relation to the utilization of the ISI as a measure. This clarification will contribute to a more accurate interpretation of our findings.

2) Reference added

Methods

  1. The choice of a cutoff score of "6" to indicate "significant sleep disorders" in our study was based on our specific research context. We recognize that the standard interpretation of the Insomnia Severity Index (ISI) suggests that a score of "6" typically indicates "no clinically significant insomnia." However, in our study we sought to capture a broader range of sleep-related difficulties, including those that may not meet the conventional threshold for clinical insomnia according to the ISI but still represent noteworthy sleep disorders assessed clinically and with the PSQI. Our decision to use a lower cutoff score is motivated by the intention to include more individuals experiencing milder forms of sleep disorders, which may be relevant to our particular research objective. 
  2. Inclusion/Exclusion Criteria Clarification:

    • We appreciate the reviewer's attention to the clarity of our inclusion/exclusion criteria. In response to the concern about the DASS (Depression, Anxiety, and Stress Scale) being used to exclude psychiatric disorders, we will revise the manuscript to explicitly state the specific psychiatric disorders we aimed to exclude. Additionally, we will provide more details on how we assessed conditions such as Bipolar disorder. Regarding the exclusion of individuals with a history of cancer in the last 5 years, we will include a rationale for this exclusion criterion to enhance transparency.

      3.  Ethical Board Requirement:

      • We acknowledge the reviewer's valid point regarding the absence of information on the ethical board requirement. But we inform that our Ministry of Health exempts us in the present case from any request for ethical approval.

Discussion

  1. Medication as the Primary Treatment vs. CBT-I:

    • We appreciate the reviewer's insight into the importance of discussing Cognitive Behavioral Therapy for Insomnia (CBT-I) as a first-line treatment. While our focus was initially on medication, we acknowledge the significance of CBT-I. In the revised manuscript, we include a section discussing the merits of CBT-I, especially in cases where medication may not be the optimal choice, such as instances of limited access to providers.
  2. Rationale for Lack of a Control Group:

    • We understand the concern regarding the absence of a control group in our study. Our rationale for this decision is rooted in population characteristics and the presence of an internal control group. However, we recognize the importance of comparison groups in research. In the revised manuscript, we will include a discussion about potential limitations arising from the lack of a control group.
  3. Discussion Scope and Literature Review:

    • The reviewer suggests narrowing down the discussion to literature directly related to our results. We acknowledge this feedback and agree that refining the discussion to focus more directly on our results is essential. In the revised manuscript, we will carefully reevaluate the discussion section, ensuring that information related to the PSQI and other details are appropriately placed in the introduction or methods sections to enhance the coherence and clarity of our presentation.

We hope that as well as the other reviewers once their suggestions are accepted you will finally allow us to publish this work that is the result of important efforts.

Best,

Matteo end Colleagues

Round 2

Reviewer 1 Report

Comments and Suggestions for Authors

Dear Authors,

Thank you for proofreading the article. However, the purpose of the work still needs to be reformulated. The goal must be precise, but without so many methodological details. For example, the average age, the number of respondents (it is enough that the research concerns adults) and the description of the procedure are not necessary, because these data are included in the methodological part.

Author Response

1) "the purpose of the work still needs to be reformulated."

We thank the reviewer for noticing this. These part is now improved in the revised version of the paper. Specifically, we have now specified that the primary goal of the study is to evaluate, through validated questionnaires, the impact of the dietary supplement SPINOFF® on sleep quality and mental well-being in a cohort of 41 adults. A secondary purpose is to assess sleep continuity through the use of a wearable smart watch.

Reviewer 2 Report

Comments and Suggestions for Authors

Dear authors,

Next time please provide a point-by-point answer (there were two similar files of the manuscript attached to this revision instead of a file with answers and the manuscript). 

You must add a short sentence to the abstract, mentioning the absence of a placebo-control. E.g. "This study assessed sleep continuity using Awakenings per Night (ApN) via a smartwatch (HELO HEALTH®) and conducted the study in two phases: baseline (T0) and after 30 days of treatment (T1) (Phase A). No placebo-control was used in this study."

Line 121 - the reference 31 is an improper citation. The original paper should be cited here (https://doi.org/10.1016/j.jep.2015.06.016).

Line 157 "thiamine acts as a co-enzyme for adenosine triphosphate (ATP) production" - It's worth mentioning the so-called non-coenzyme action of thiamine here with some examples of molecular targets. 

Lines 269-271. Although using t-test is OK for the phase A, because the number of individuals is rather big, the phase B lacks confirmation of the normality testing. Please confirm that the data parameters were distributed normally, especially for the phase B. Mention the test used.

Mention the n= in tables 2 and 3, similarly to table 1.

Lines 399-400. "This study did not include a control group because in our opinion it would have been unethical to leave some of these patients, who reported sleep disorders, without any treatment". In this regard you should extend the discussion here by mentioning the possible placebo effect, which is indistinguishable from the effect of the drug by design.

Since a daytime-dependent effect of thiamine has been reported, there is a possibility, the effect of the drug depends on the time of its supplementation. This is true at least for the vitamin B1 (DOI: 10.1023/a:1023276009477  and https://doi.org/10.1111/jnc.14951).

Typos (e.g. line 25) should be checked.

Conflict of interest. It's better to state here again, that the two companies had no role in the design of the study; in the collection, analyses, or interpretation of data; in the writing of the manuscript; or in the decision to publish the results. Otherwise someone might see a conflict of interest.

Author Response

Dear reviewer,

thank you for your clarification.
We apologize if the responses in the first round of review were non-point-by-point, as we accidentally did not attach the correct file. 

1)"You must add a short sentence to the abstract, mentioning the absence of a placebo-control"

We thank the reviewer for noticing this. These part is now improved as you suggest.

2) "the reference 31 is an improper citation"

We thank the reviewer for noticing this. These part is now improved in the revised version of the paper.

3) Line 121 - the reference 31 is an improper citation.

We thank the reviewer for noticing this. These part is now improved in the revised version of the paper.

4) We thank the reviewer for this remark. Normality of outcome variables in phase B was assessed and verified (p=0.084 and p=0.126, respectively) using Shapiro-Wilks test. This detail was also added in statistical analysis section.

5) Information on “n” was added in table 2 and 3.

6) "you should extend the discussion here by mentioning the possible placebo effect, which is indistinguishable from the effect of the drug by design."

We thank the reviewer for noticing this. These part is now improved as you suggest. "The absence of a control group, driven by ethical considerations, has made it challenging to distinguish the true therapeutic effects of the drug from the placebo response. While our findings indicate positive outcomes, it's essential to acknowledge the inherent limitations in interpreting these results and the need for further research that incorporates control groups to better understand the true efficacy of the treatment. This consideration highlights the delicate balance between ethical concerns and scientific rigor in clinical research."

7) "Since a daytime-dependent effect"

Thank you for noting this, we mentioned this effect in the discussion by placing these works as a reference.

8) "Typos"

We thank the reviewer for noticing this. These part is now improved as you suggest.

9) "Conflict of interest"

We thank the reviewer for noticing this. These part is now improved as you suggest.

Reviewer 3 Report

Comments and Suggestions for Authors

Thank you for the thorough and prompt response to review. All issues have been addressed. 

Round 3

Reviewer 2 Report

Comments and Suggestions for Authors

The authors have answered all my previous questions and comments.

I still have to focus attention of the Editors to the absence of placebo-control in this study, which, according to the authors, is explained by the ethical reasons. This fact is always properly mentioned by them, so no misleading interpretations should occur. With this in mind I believe, the manuscript may be published after minor corrections.

I suggest to reformulate the goal, because the sentence "A secondary purpose is to assess sleep continuity through the use of an wearable smart watch" doesn't suit as a goal. In my opinion, using the watch is just a modern method, which is very suitable for the study under consideration.

Author Response

Dear Reviewer,
Thank you for noticing and for the suggestion, as you will see on line 105, the secondary purpose of the study has been modified and made unambiguous in its interpretation.